# Passive Aggressive Ensemble for Online Portfolio Selection

**Kailin Xie [1], Jianfei Yin [1], Hengyong Yu [2], Hong Fu [3,\*] and Ying Chu [1,\*]**

[1] College of Computer Science and Software Engineering, Shenzhen University, Shenzhen 518060, China; 2110276045@email.szu.edu.cn (K.X.); yjf@szu.edu.cn (J.Y.)
[2] Department of Electrical and Computer Engineering, University of Massachusetts Lowell, Lowell, MA 01854, USA; hengyong_yu@uml.edu
[3] Department of Mathematics and Information Technology, The Education University of Hong Kong, Hong Kong, China
[\*] Correspondence: hfu@eduhk.hk (H.F.); chuying@szu.edu.cn (Y.C.)

**Abstract:** Developing effective trend estimators is the main method to solve the online portfolio selection problem. Although the existing portfolio strategies have demonstrated good performance through the development of various trend estimators, it is still challenging to determine in advance which estimator will yield the maximum final cumulative wealth in online portfolio selection tasks. This paper studies an online ensemble approach for online portfolio selection by leveraging the strengths of multiple trend estimators. Specifically, a return-based loss function and a cross-entropy-based loss function are first designed to evaluate the adaptiveness of different trend estimators in a financial environment. On this basis, a passive aggressive ensemble model is proposed to weigh these trend estimators within a unit simplex according to their adaptiveness. Extensive experiments are conducted on benchmark datasets from various real-world stock markets to evaluate their performance. The results show that the proposed strategy achieves state-of-the-art performance, including efficiency and cumulative return.

**Keywords:** online portfolio selection; online ensemble learning; passive aggressive algorithm

**MSC:** 68T09

## 1. Introduction

Online portfolio selection dynamically allocates wealth across real-world assets by analyzing stream data from financial markets [1,2]. To optimize wealth allocation, portfolio selection leverages a range of machine learning techniques, including the Newton gradient step [3] and nearest neighbor [4]. In addition to these methods, portfolio selection must also consider economic and financial theories and criteria, particularly investment behaviors [5,6]. Accordingly, many of the current leading portfolio selection systems employ historical price information to predict future prices to determine the optimal portfolio allocation.

Reversion and momentum are two significant characteristics in finance. Numerous state-of-the-art portfolio selection strategies develop trend estimators based on these characteristics for decision-making. For example, trend estimators like the inverse price [7,8], the simple moving average, and the exponential moving average [9] assume that the asset price will reverse to some kind of historical average. On the contrary, the peak price [10] assumes that the price of a well-performing asset will continue to rise due to irrational investing behaviors. Since each trend estimator is based on different prior assumptions, their effectiveness is contingent on the financial environment aligning with these assumptions. Furthermore, existing methods [11,12] attempt to utilize multiple trend estimators, but as the number of trends and assets increases, the computational cost becomes very high due to matrix calculations. It is still challenging to predetermine which trend estimator will maximize the final cumulative wealth efficiently for online portfolio selection tasks.

To take advantage of various trend estimators during the whole investment period efficiently, we pay attention to the online ensemble learning framework [13]. The online learning adaptively balances each trend estimator with evolving financial environments, while the ensemble learning combines these estimators to enhance robustness and precision. A simple but effective ensemble learning approach is weighted averaging [14]. Consequently, developing a suitable online ensemble learning algorithm to amalgamate various trend estimators is a viable solution.

In this paper, we address the problem of online portfolio selection with multiple trend estimators by developing a framework named passive aggressive ensemble (PAE). The characteristics and contributions of the proposed PAE are summarized as follows:

1. The PAE framework is introduced to employ two distinct schemes to efficiently ensemble four different types of trend estimators with a lower time complexity of $O(Lmn)$, where $m$ is the number of assets, $L$ is the number of trends, and $n$ is the total investment period.
2. The PAE framework augments the performance of original trend estimators through reasonable evaluation and weighting.
3. Extensive experiments are conducted on six real-world datasets to demonstrate that our algorithm not only outperforms competing strategies in terms of multiple evaluation criteria but also has promising scalability in terms of transaction costs.

The rest of the paper is organized as follows. Section 2 reviews some related work regarding trend estimators and ensemble learning in the context of online portfolio selection and formally states portfolio selection in machine learning. Section 3 presents our PAE framework for portfolio selection. Section 4 presents the experiments and results of PAE on real-world benchmark datasets. Finally, conclusions are made in Section 5.

## 2. Related Works and Problem Setting

### 2.1. Related Works

#### 2.1.1. Trend Estimator

Following-the-winner strategies assume that the price of a good performing asset will continue to rise. Based on this assumption, researchers have designed different trend estimators. Exponential gradient [15] and its improved variants [16] estimate the asset price as it will continue to change as of last day. The peak price tracking system (PPT) [10] and short-term portfolio optimization with loss control [17] adopt the peak price (PP) as a trend estimator. PP assumes the price continues to move at its maximum price in a recent time window but fails to capture the continuous depreciation of the asset price. Recently, there have been many works to handle the shortcomings of PP, for example, the trend peak price tracing [11] and the peak price tracking approach [18].

Following-the-loser strategies are opposite to following-the-winner strategies, and they assume that the asset price will reverse to some kind of historical average. The inverse price (IP) expects the asset price to reverse on the last day. Passive aggressive mean reversion [7] and confidence weighted mean reversion [8] adopt the IP as a trend estimator but suffer from significant performance degradation if the underlying short-term reversion fails to exist. On-Line Moving Average Reversion (OLMAR) [9] uses the simple moving average (SMA) and exponential moving average (EMA) to predict the next prices. The SMA and EMA improve the estimation of IP. The SMA utilizes the average of the asset prices within a specified time window to estimate the future price movement. It assigns equal weight to each asset price in the specified period, effectively smoothing out any extreme prices within the time frame to derive the most recent price information from the financial market. The EMA incorporates all historical asset prices but assigns higher significance to recent prices in predicting future trends, making it generally more reflective of the actual price trends in comparison to the SMA. Recently, Gaussian weighting reversion [19] has used the Gaussian weighting function to enhance time validity. Vector autoregressive weighting reversion [20] and weighted multivariate mean reversion [21] use autoregressive moving average and multi-variate robust mean, respectively, to estimate the mean more

precisely. The state-dependent EMA [22] predicts asset returns by additionally analyzing market states.

### 2.1.2. Ensemble Learning

In recent years, ensemble learning algorithms have gained wider recognition for their enhanced generalization ability and broader application scope. The online portfolio selection has garnered significant interest from both industry professionals and academic researchers. In this context, Lai et al. [12] proposed an adaptive input and composite trend representation (AICTR) to fuse three trends. It used radial basis functions to fuse three trends (SMA, EMA, PP) by considering both the adaptive input and each trend estimator and reaching a time complexity of $O((m + L)mn)$, where $m$ is the number of assets, $L$ is the number of trends, and $n$ is total investment period. Guo et al. [23] enhanced the prediction of future returns of volatile assets by integrating an adaptive decaying factor with the traditional moving average approach but did not analyze the time complexity. Kumar and Segev [24] employed Bayesian methods to ensemble two strategies but not a trend estimator, improving future portfolio decisions by learning from past prediction errors. Dai et al. [11] introduced trend peak price tracing (TPPT), a dynamic price forecasting method that adapts to market conditions by incorporating PP, EMA, or maintaining current values based on trend analysis, but it reaches a time complexity of $O(m^2 n)$.

### 2.2. Problem Setting

In this paper, we use a standard and universal setting for portfolio selection in machine learning [7–12]. Consider a financial market with $m$ assets for $n$ periods. At the end of the $t$th period, a non-negative $m$-dimensional vector as follows:

$$\mathbf{p}_t \in \mathbb{R}_+^m, \quad t = 0, 1, \ldots, n \ , \tag{1}$$

represents the close price of assets. A relative price vector is introduced to see the change of asset prices as follows:

$$\mathbf{x}_t \triangleq \frac{\mathbf{p}_t}{\mathbf{p}_{t-1}}, \quad \mathbf{x}_t \in \mathbb{R}_+^m, \qquad t = 1, 2, \ldots, n, \tag{2}$$

where a division between two vectors represents element-wise division in this paper.

At the beginning of the $t$th period, an investment in the market is specified by a portfolio vector in $m$ dimensional unit simplex, as follows:

$$\mathbf{b}_t \in \Delta_m := \left\{ \mathbf{b} \in \mathbb{R}_+^m : \sum_{i=1}^{m} \mathbf{b}^{(i)} = 1 \right\}, \tag{3}$$

where $\mathbf{b}_t^{(i)}$ denotes the proportion of total wealth invested in the $i$th asset. The non-negative constrain means no short is allowed, and the equality constraint means that the portfolio is self-financed.

For the $t$th trading day, a portfolio $\mathbf{b}_t$ generated by the portfolio selection algorithm results in a daily return $\mathbf{b}_t^\top \mathbf{x}_t$. Thus, the cumulative wealth can be calculated as $S_t = S_{t-1} \cdot \left( \mathbf{b}_t^\top \mathbf{x}_t \right)$, and the final cumulative wealth with a common initial wealth $S_0 = 1$ is given by the following:

$$S_n = S_0 \prod_{t=1}^{n} \left( \mathbf{b}_t^\top \mathbf{x}_t \right) = \prod_{t=1}^{n} \left( \mathbf{b}_t^\top \mathbf{x}_t \right). \tag{4}$$

Finally, a portfolio learning algorithm sequentially learns a set of portfolio vectors $\{\mathbf{b}_t\}_{t=1}^{n}$ to maximize the final cumulative wealth and satisfy some risk management metrics.

## 3. Methodology

In this section, we propose a passive aggressive ensemble (PAE) framework and two strategies, PAE-R and PAE-C, to take advantage of different trends and adapt to various financial environments. We take four traditional trend estimators as the input for PAE:

$$\text{SMA}: \quad \hat{\mathbf{x}}_{S,t+1}(w) = \frac{\sum_{k=0}^{w-1} \mathbf{p}_{t-k}}{w \mathbf{p}_t}, \tag{5}$$

$$\text{EMA}: \quad \hat{\mathbf{x}}_{E,t+1}(\vartheta) = \vartheta \mathbf{1} + (1 - \vartheta) \frac{\hat{\mathbf{x}}_{E,t}}{\mathbf{x}_t}, \tag{6}$$

$$\text{IP}: \quad \hat{\mathbf{x}}_{I,t+1} = \frac{1}{\mathbf{x}_t} = \frac{\mathbf{p}_{t-1}}{\mathbf{p}_t}, \tag{7}$$

$$\text{PP}: \quad \hat{\mathbf{x}}_{M,t+1} = \frac{\hat{\mathbf{p}}_{t+1}}{\mathbf{p}_t}, \quad \hat{\mathbf{p}}_{t+1}^{(i)} = \max_{0 \le k \le w-1} \mathbf{p}_{t-k}^{(i)}, \tag{8}$$

where $w$ is the time window size, $\vartheta \in (0,1)$ denotes the decaying factor, $\mathbf{1}$ is a $m$-dimensional vector whose elements are 1, and $i = 1, 2, \ldots, m$ is the order of the asset.

### 3.1. Passive Aggressive Ensemble

The construction of trend composites starts with a trend back-test step. Let $\{\hat{\mathbf{x}}_{l,t+1}\}_{l=1}^{L}$ denote a set of trends—for example, $\hat{\mathbf{x}}_{1,t+1} = \hat{\mathbf{x}}_{S,t+1}$, $\hat{\mathbf{x}}_{2,t+1} = \hat{\mathbf{x}}_{E,t+1}$ and $\hat{\mathbf{x}}_{3,t+1} = \hat{\mathbf{x}}_{I,t+1}$, where $L$ is the total number of trends. We project $\{\hat{\mathbf{x}}_{l,t+1}\}_{l=1}^{L}$ onto a unit simplex [25] to obtain feasible portfolios as follows:

$$\widetilde{\mathbf{x}}_{l,t+1} = \operatorname*{argmin}_{\mathbf{x} \in \Delta_m} \| \mathbf{x} - \hat{\mathbf{x}}_{l,t+1} \|^2, \quad l = 1, \ldots, L. \tag{9}$$

Then we are able to measure the return-based performance [1] of each trend estimator in a certain time period as follows:

$$r_{l,t} = \widetilde{\mathbf{x}}_{l,t}^{\top} \mathbf{x}_t, \qquad \mathbf{r}_t = [r_{1,t}, r_{2,t}, \ldots, r_{L,t}]^{\top}, \tag{10}$$

where $r_{l,t}$ is the back-tested return of $l$th trend in $t$th period, $\mathbf{x}_t$ is the real relative price factor given by (2), and $\mathbf{r}_t \in \mathbb{R}^L$ is the collection of $L$ different back-tested returns.

Drawing upon from the cross-entropy [26], we also propose an indicator to evaluate each trend estimator called cross-entropy-based performance as follows:

$$c_{l,t} = -\widetilde{\mathbf{x}}_t^{\top} \log \widetilde{\mathbf{x}}_{l,t}, \quad \mathbf{c}_t = [c_{1,t}, c_{2,t}, \ldots, c_{L,t}]^{\top}, \tag{11}$$

where $\widetilde{\mathbf{x}}_t$ is the projection of the real relative price vector by (9). The cross-entropy allows us to measure the gap between the trend estimator and the true price movement.

Two ensemble targets are defined to meet the goal of our approach. We aim to ensure that our ensembled trend estimator is at least as effective as the average performance of any single trend estimator over the investment period, aligning with investment intuition. Smaller values for cross-entropy-based performance indicate a better trend estimator, while the opposite holds true for returns-based performance. Our ensemble targets are defined as follows:

$$r_{*,t} \triangleq \operatorname*{argmax}_{1 \le l \le L} \frac{1}{w} \sum_{k=0}^{w-1} r_{l,t-k}, \tag{12}$$

$$c_{*,t} \triangleq \operatorname*{argmin}_{1 \le l \le L} \frac{1}{w} \sum_{k=0}^{w-1} c_{l,t-k}, \tag{13}$$

where $w$ is the window size in the most recent time. We average the back-tested performance of each trend estimator in the most recent time window and select the one with the best average performance.

Combining the passive aggressive principle [27] and online ensemble framework [13], a weighting vector $\mathbf{w}_t \in \Delta_L$ for weighted averaging is first defined. Then, we propose two new loss functions to combine all the trends and consider their performance for our PAE-R and PAE-C strategies, respectively:

$$\text{PAE-R}: \ell_{\xi_r}(\mathbf{w}; (\mathbf{r}_t, r_{*,t})) = \begin{cases} 0 & r_{*,t} - \mathbf{w}^\top \mathbf{r}_t \leq \xi_r \\ r_{*,t} - \mathbf{w}^\top \mathbf{r}_t - \xi_r & \text{otherwise} \end{cases}, \tag{14}$$

$$\text{PAE-C}: \ell_{\xi_c}(\mathbf{w}; (\mathbf{c}_t, c_{*,t})) = \begin{cases} 0 & \mathbf{w}^\top \mathbf{c}_t - c_{*,t} \leq \xi_c \\ \mathbf{w}^\top \mathbf{c}_t - c_{*,t} - \xi_c & \text{otherwise} \end{cases}, \tag{15}$$

where $r_{*,t}$ and $c_{*,t}$ are the performance of ensemble targets, $\xi_r$ is the tolerance threshold between the return-based performance of ensemble target and weighted averaging trend estimators, and $\xi_c$ is the tolerance threshold between the cross-entropy-based performance of ensemble target and weighted averaging trend estimators.

We update the weighting vector at the end of each trading period through the following optimization problem as follows:

$$\mathbf{w}_{t+1} = \underset{\mathbf{w} \in \Delta_L}{\operatorname{argmin}} \frac{1}{2} \| \mathbf{w} - \mathbf{w}_t \|^2 \quad \text{s.t.} \begin{cases} \ell_{\xi_r}(\mathbf{w}; (\mathbf{r}_t, r_{*,t})) = 0 & \text{PAE-R} \\ \ell_{\xi_c}(\mathbf{w}; (\mathbf{c}_t, c_{*,t})) = 0 & \text{PAE-C} \end{cases}. \tag{16}$$

On one hand, if our weighted averaging trend estimators outperform or are close to the ensemble target, our approach passively maintains the weighting vector $\mathbf{w}$ in the next trading period. On the other hand, when the weighted averaging trend estimators perform much worse than the ensemble target, our approach aggressively adjusts $\mathbf{w}$ to seek a better weight. The problem will be solved in Section 4.3.

### 3.2. Online Portfolio Selection with Multiple Trend Estimators

The next step is to decide the proportion of the wealth being allocated to each asset of the stock based on the weighting and trend estimators received from the previous step. First proposed by Li et al. [7,9,27], portfolio selection through passive aggressive algorithms is widely used today [19,21,24], especially in single trend estimator situations. We extend the application of this algorithm to the case of multiple trend estimators. Our basic idea to form the final portfolio is to maximize the ensemble trend estimator and keep the last portfolio information through a regularization term. The final portfolio optimization problem can be expressed as follows:

$$\mathbf{b}_{t+1} = \underset{\mathbf{b} \in \Delta_m}{\operatorname{argmin}} \frac{1}{2} \| \mathbf{b} - \mathbf{b}_t \|^2 \quad \text{s.t.} \ \mathbf{b}^\top \hat{\mathbf{X}}_{t+1} \mathbf{w}_{t+1} \geq \epsilon, \ \hat{\mathbf{X}}_{t+1} = [\hat{\mathbf{x}}_{1,t+1}, \ldots, \hat{\mathbf{x}}_{L,t+1}], \tag{17}$$

where $\hat{\mathbf{X}}_{t+1} \in \mathbb{R}^{m \times L}$ is the collection of $L$ different trends, $\mathbf{w}_{t+1} \in \Delta_L$ is the weighting vector, $\epsilon$ is the reversion threshold, and $\hat{\mathbf{X}}_{t+1} \mathbf{w}_{t+1}$ is the ensemble trend estimator.

If the expected return $\mathbf{b}^\top \hat{\mathbf{X}}_{t+1} \mathbf{w}_{t+1}$ is greater than the threshold, we passively maintain the original investment portfolio vector. If the expected return is less than the threshold, we aggressively adjust the investment portfolio vector to maximize our returns. $\hat{\mathbf{X}}_{t+1} \mathbf{w}_{t+1}$ is capable of combining various trends via the weighting vector $\mathbf{w}_{t+1}$ as the output of (16). We will solve this problem in Section 4.3.

### 3.3. Solving Algorithm

In this subsection, we will derive the solutions for the above optimization problem and summarize the whole procedure for PAE algorithms in Algorithm 1.

**Theorem 1.** *Without considering the non-negativity constraint, the solution of the PAE-R type optimization problem (16) is as follows:*

$$\mathbf{w}_{t+1} = \mathbf{w}_t + \tau_{1,t}(\mathbf{r}_t - \bar{r}_t\mathbf{1}), \tag{18}$$

*where* $\bar{r}_t = \frac{\mathbf{1}^\top \mathbf{r}_t}{L}$ *is the mean value of back-tested return-based performance and* $\tau_{1,t}$ *is computed as follows:*

$$\tau_{1,t} = \max\left\{0, \frac{r_{*,t} - \mathbf{w}^\top \mathbf{r}_t - \xi_r}{\|\mathbf{r}_t - \bar{r}_t\mathbf{1}\|^2}\right\}. \tag{19}$$

The proof of this theorem is in Appendix A.

**Theorem 2.** *Without considering the non-negativity constraint, the solution of the PAE-C type optimization problem (16) is as follows:*

$$\mathbf{w}_{t+1} = \mathbf{w}_t + \tau_{1,t}(\bar{c}_t\mathbf{1} - \mathbf{c}_t), \tag{20}$$

*where* $\bar{c}_t = \frac{\mathbf{1}^\top \mathbf{c}_t}{L}$ *is the mean value of the back-tested cross-entropy-based performance and* $\tau_{1,t}$ *is computed as follows:*

$$\tau_{1,t} = \max\left\{0, \frac{\mathbf{w}^\top \mathbf{c}_t - c_{*,t} - \xi_c}{\|\bar{c}_t\mathbf{1} - \mathbf{c}_t\|^2}\right\}. \tag{21}$$

Because the proof of Theorem 2 is almost the same as Theorem 1, we omit the proof.

No matter which type of loss function is considered, $\mathbf{w}_{t+1}$ should be further projected onto the $L$-dimensional unit simplex [25] to satisfy the non-negativity constraint of weighted averaging as follows:

$$\mathbf{w}_{t+1} = \underset{\mathbf{w}\in\Delta_L}{\arg\min}\| \mathbf{w} - \mathbf{w}_{t+1} \|^2. \tag{22}$$

**Theorem 3.** *Without considering the non-negativity constraint, the solution of problem (17) is as follows:*

$$\mathbf{b}_{t+1} = \mathbf{b}_t + \tau_{2,t}\left(\mathbf{b}_t - \mu\mathbf{1}_{(m)}\right), \tag{23}$$

*where* $\mathbf{1}_{(m)}$ *denotes the m-dimensional* **1** *vector to distinguish with L-dimensional* **1** *in (19) and (21), and* $\mu = \frac{1}{m}\mathbf{1}_{(m)}^\top \hat{\mathbf{X}}_{t+1}\mathbf{w}_{t+1}$ *is the average value of trend composite. Then,* $\tau_{2,t}$ *is calculated as follows:*

$$\tau_{2,t} = \max\left\{0, \frac{\epsilon - \mathbf{b}_t^\top \hat{\mathbf{X}}_{t+1}\mathbf{w}_{t+1}}{\|\hat{\mathbf{X}}_{t+1}\mathbf{w}_{t+1} - \mu\mathbf{1}_{(m)}\|^2}\right\}. \tag{24}$$

The proof of Theorem 3 is in Appendix B.

$\mathbf{b}_{t+1}$ is further projected onto a $m$-dimensional unit simplex [25] to satisfy the non-negativity constraint as follows:

$$\mathbf{b}_{t+1} = \underset{\mathbf{b}\in\Delta_m}{\arg\min}\| \mathbf{b} - \mathbf{b}_{t+1} \|^2. \tag{25}$$

Algorithm 1 summarizes the general computing process of PAE-R and PAE-C.

---

**Algorithm 1** PAE framework

---

**Input:**  Given parameters $w$, $\xi_r$, $\xi_c$, $\epsilon$, the trends $\{\hat{\mathbf{x}}_{l,t-k}\}_{k=-1}^{w-1}$ and the actual price relatives $\{\mathbf{x}_{t-k}\}_{k=0}^{w-1}$ in recent time window, the current weighting vector $\mathbf{w}_t$ and the current portfolio $\mathbf{b}_t$.

**Output:**  The next portfolio $\mathbf{b}_{t+1}$.

1 Calculating the feasible portfolio $\{\tilde{\mathbf{x}}_{l,t-k}\}_{k=0}^{w-1}$ by (9);

2 Calculating the recent back-tested returns $\{r_{l,t-w}\}_{k=0}^{w-1}$ by (10) and $\{c_{l,t-w}\}_{k=0}^{w-1}$ by (11) to measure the performance of each trend;

3 Defining the ensemble target $r_{*,t}$ by (12) and $c_{*,t}$ by (13);

4 Setting parameter $\tau_{1,t}$ :

$$\tau_{1,t} = \begin{cases} \max\left\{0, \dfrac{r_{*,t}-\mathbf{w}^\top\mathbf{r}_t-\xi_r}{\|\mathbf{r}_t-\bar{r}_t\mathbf{1}\|^2}\right\} & \text{PAE-R} \\[2ex] \max\left\{0, \dfrac{\mathbf{w}^\top\mathbf{c}_t-c_{*,t}-\xi_c}{\|\bar{c}_t\mathbf{1}-\mathbf{c}_t\|^2}\right\} & \text{PAE-C} \end{cases} ;$$

5 Updating weighting vector:

$$\mathbf{w}_{t+1} = \begin{cases} \mathbf{w}_t + \tau_{1,t}(\mathbf{r}_t - \bar{r}_t\mathbf{1}) & \text{PAE-R} \\ \mathbf{w}_t + \tau_{1,t}(\bar{c}_t\mathbf{1} - \mathbf{c}_t) & \text{PAE-C} \end{cases} ;$$

6 Normalizing the next weighting vector $\mathbf{w}_{t+1}$ by (22);

7 Setting parameter $\tau_{2,t}$ by (24);

8 Updating $\mathbf{b}_{t+1} = \mathbf{b}_t + \tau_{2,t}\left(\mathbf{b}_t - \mu\mathbf{1}_{(m)}\right)$;

9 Normalizing the next portfolio $\mathbf{b}_{t+1}$ by (25).

---

### 3.4. Complexity Analysis

In the context of online portfolio selection, running efficiency is a crucial factor to consider as the application has time constraints. The ideal candidate for high-frequency trading [28] would be a portfolio selection strategy that effectively balances wealth accumulation and computational efficiency. The main factors that affect the computational cost of PAE are the total investing periods $n$, the number of assets $m$, and the number of trends $L$. All the steps of PAE cost $O(m)$ except step 7, which consumes $O(Lm)$. Table 1 shows the time complexity for different portfolio selection strategies. Since $L$ is usually much smaller than d, our PAE algorithm achieves the lowest complexity when using multiple trend estimators. As the number of trends and assets increases, PAE is still efficient to compute. To be specific, PAE maintains the complexity of $O(Lmn)$, which makes it applicable to time-limited applications.

**Table 1.** Complexity comparison with other strategies.

| Type | Strategies | Complexity |
|:---:|:---:|:---:|
| Single trend estimator | OLMAR | $O(mn)$ |
| | PPT | $O(m^2n)$ |
| Multiple trend estimators | AICTR | $O((m+L)mn)$ |
| | TPPT | $O(m^2n)$ |
| | PAE | $O(Lmn)$ |

## 4. Experiments and Results

In this section, we mainly focus on the comparison studies. First, we introduce experimental datasets and competing portfolio strategies and criteria of evaluation in turn. Then, we conduct experiments on parameter settings and ensemble effectiveness. Finally, we report and analyze the results of comparison studies and also discuss transaction costs and running time.

*4.1. Data*

To improve the reproducibility of the experiments and get close to the recent market, we conducted comparison studies on four popular traditional datasets and three new datasets. All of them are publicly available. Three traditional datasets include NYSE(N) [8], TSE [29], and MSCI [7]. Three new datasets include NYSE19 [17], ZZ28 [30], and ETF23.

They contain real-world daily close price relatives from different stock and index markets, including the New York Stock Exchange (NYSE(N), NYSE19), the Toronto Stock Exchange (TSE), the MSCI World Index (MSCI), and the China Securities 500 index (ZZ28). In 2023, China's ETF market experienced significant development, with a total scale exceeding 1.8 trillion yuan, marking a nearly 40% increase from 2022. The remarkable growth of this market underscores its importance, leading us to propose the ETF23 dataset. This dataset includes the largest ETFs by market capitalization from 23 different industries in the Chinese ETF market. Detailed information about these datasets is provided in Table 2.

**Table 2.** Detailed information for six datasets.

| Dataset | Region | Time | Days | Assets |
|---------|--------|------|------|--------|
| NYSE(N) | US | 1 January 1985–30 June 2010 | 6431 | 23 |
| TSE | CA | 1 January 1994–31 December 1998 | 1259 | 88 |
| MSCI | Global | 1 April 2006–31 March 2010 | 1043 | 24 |
| NYSE19 | US | 2 January 2015–4 September 2019 | 1167 | 47 |
| ZZ28 | CN | 4 January 2000–1 April 2020 | 4905 | 28 |
| ETF23 | CN | 1 February 2021–1 October 2023 | 647 | 23 |

*4.2. Competing Portfolio Strategies*

For comparison, we consider some benchmarks and a number of existing online portfolio management strategies (including some state-of-the-art ones). In the following, we show these benchmarks and strategies, where the parameters of each strategy are set according to the recommendations of the corresponding studies.

- BAH: the uniform Buy-And-Hold trading strategy. The strategy invests equally in $m$ assets at the onset and maintains this allocation throughout.
- OLMAR [9]: It takes the moving average to predict the future price. The parameters are set as follows: $w = 5$ and $\epsilon = 10$.
- AICTR [12]: It combines three trends (SMA, EMA, PP) and market conditions through radial basis functions. The parameters are set as follows: $w = 5$, $\sigma_l^2 = 0.0025$, and $\epsilon = 1000$.
- SPOLC [17]: The short-term portfolio optimization with loss control strategy with the window size $w = 5$ and the mixing parameter $\gamma = 0.025$.
- TPPT [11]: It uses adjustable historical windows and slope values for price prediction. The parameters are set as follows: $w = 5$, $\alpha = 0.5$, and $\epsilon = 100$.

*4.3. Evaluation Criteria*

For ease of calculation, we assume that the initial wealth is 1 [31]. Under such a circumstance, the single-period growth rate equals the portfolio's cumulative return at the $t$th period, and the cumulative growth rate is equal to cumulative wealth. Then, we evaluate the characteristics of the aforementioned portfolio strategies using two criteria and five performance metrics. The two criteria are return and risk-adjusted return. Basically, the higher the values of metrics in return and risk-adjusted return criteria, the better the portfolio strategy performs.

- Cumulative Wealth (CW). The CW serves as the principal metric for evaluating the investment performance of each portfolio selection algorithm. The CW is computed by (4).

- Annualized Percentage Yield (APY). The APY is a widely used metric for evaluating investment returns. It represents the average return of a strategy over the course of a year. APY is computed as follows:

$$\text{APY} = S_n^{1/y} - 1, \tag{26}$$

where $y$ represents the number of years according to $n$ trading days. In this study, all datasets consist of daily prices. Therefore, $y$ is calculated as $n$ divided by 252, which is the average number of annual trading days.

- Sharpe Ratio (SR). In the realm of financial trading, it is often observed that higher returns are accompanied by elevated levels of risk. Thus, it is crucial for an investment algorithm to strike a balance between maximizing returns and managing risks. The SR serves as a widely utilized metric for evaluating risk-adjusted returns and is defined as follows:

$$\text{SR} = \frac{\bar{r}_s - r_f}{\sigma(r_s)}, \tag{27}$$

where $\bar{r}_s$ is the mean of $r_s$, $r_f$ is the return of a risk-free asset and is set to 0 in this paper since we do not consider a risk-free asset, and $\sigma(r_s)$ is the standard deviation of return $r_s$ estimated by the samples $r_{s,t}$ in $n$ trading periods.

- Information Ratio (IR) [32,33]. The IR is a performance evaluation metric that quantifies the excess risk-adjusted return of an investment strategy compared to a benchmark. It is defined as follows:

$$\text{IR} = \frac{\bar{r}_s - \bar{r}_m}{\sigma(r_s - r_m)}. \tag{28}$$

- Calmar Ratio (CR) [34]. The CR is a comparison of the average annual compound return and the maximum drawdown (MDD) risk, which is widely adopted in fund management. The calculation formula is CR = APY/MDD, where MDD = $\max\limits_{t \in [1,T]} \frac{M_t - S_t}{M_t}$, $M_t = \max\limits_{k \in [1,t]} S_k$.

In addition, to test whether the strategy achieves its returns just by luck [32], we introduce the statistic $t$-test method into the measurement of the proposed strategies. Particularly, a regression model of portfolio excess returns and market excess returns is established as follows:

$$r_{s,t} = \alpha + \beta r_{m,t} \ . \tag{29}$$

By assuming that parameter $\alpha$ follows a normal distribution, we conduct a statistical $t$-test on $\alpha$ and derive the probability of achieving the excess return by luck.

*4.4. Results*

4.4.1. Parameter Setting

We conduct a comprehensive analysis of the parameter setting of PAE through experiments using benchmark data sets. Similar to previous studies [9,17,19,35,36], we adopt an empirical method to determine the parameters based on their CW computed by (4).

We take four trends ($\hat{\mathbf{x}}_{S,t+1}$, $\hat{\mathbf{x}}_{E,t+1}$, $\hat{\mathbf{x}}_{I,t+1}$ and $\hat{\mathbf{x}}_{P,t+1}$) as the input of PAE and initiate the weighting vector as $\mathbf{w}$ = [1/4, 1/4, 1/4, 1/4]. The value of $w$ is set to 5. It aligns with previous research [9–12,17] and is a commonly used time window size in stock and futures investment because it reflects the recent financial environment. To run the trend back-test step, the PAE strategies require at least $w + 1$ days of data. For fairness, all strategies adopt their respective algorithm outputs for investment from the sixth day.

Regarding the threshold parameters $\xi_r$, $\xi_c$ and $\epsilon$, an initial approximation is made, followed by fine-tuning in incremental steps. For PAE-R, a fixed value of $\xi_r = 6 \times 10^{-4}$ is determined, with $\epsilon$ being varied between 10 and 110. The results shown in Figure 1 indicate that PAE-R is stable at $\epsilon = 30$. Subsequently, a fixed value of $\epsilon = 30$ is determined, with $\xi_r$ being varied between $5.5 \times 10^{-4}$ and $7.5 \times 10^{-4}$. The results shown in Figure 2 indicate

that PAE-R is stable at $\xi_r = 6 \times 10^{-4}$. For PAE-C, we repeat the above process and show the experimental results in Figures 3 and 4. From the experimental results, we recommend choosing $\epsilon = 30$ and $\xi_r = 6 \times 10^{-4}$ for PAE-R, $\epsilon = 30$ and $\xi_c = 1.5$ for PAE-C. The rest of the experiments adopt this parameter setting for comparison.

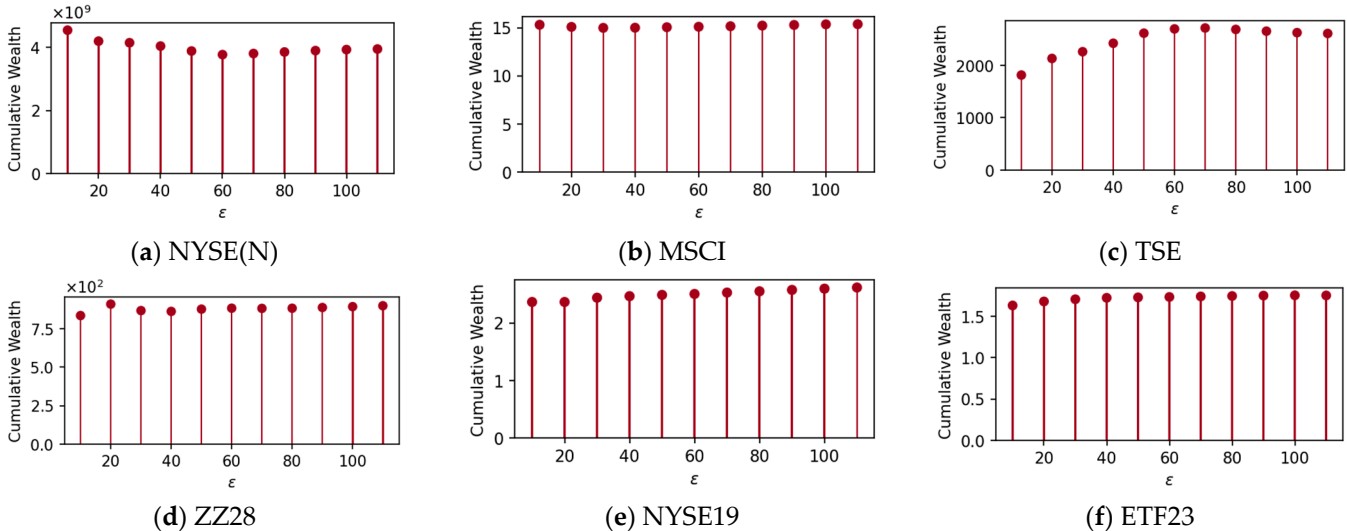

**Figure 1.** Fixed $\xi_r = 6 \times 10^{-4}$, the cumulative wealth changes with the variation of $\epsilon$ on six datasets for PAE-R.

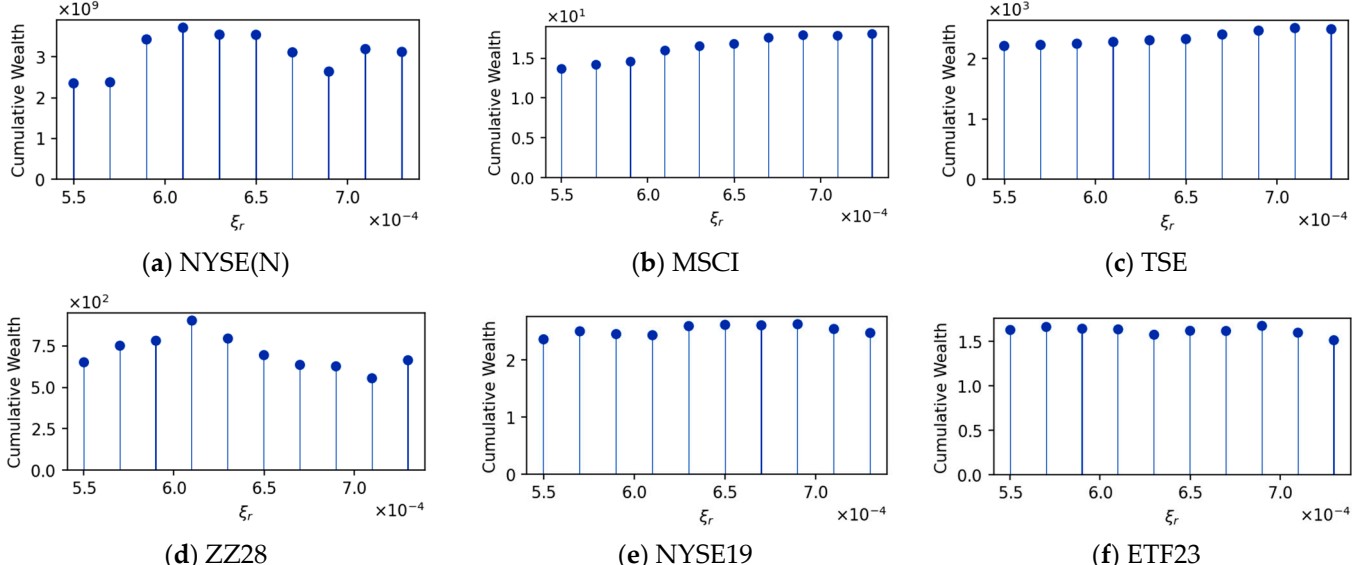

**Figure 2.** Fixed $\epsilon = 30$, the cumulative wealth changes with the variation of $\xi_r$ on six datasets for PAE-R.

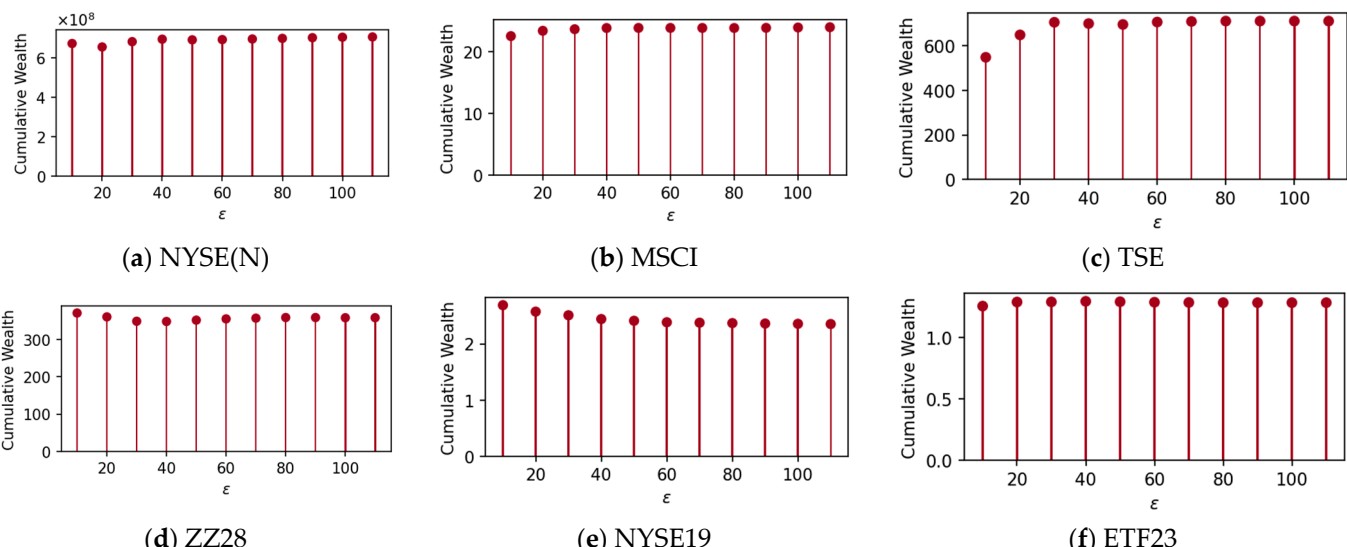

**Figure 3.** Fixed $\xi_c = 1.5$, the cumulative wealth changes with the variation of $\epsilon$ on six datasets for PAE-C.

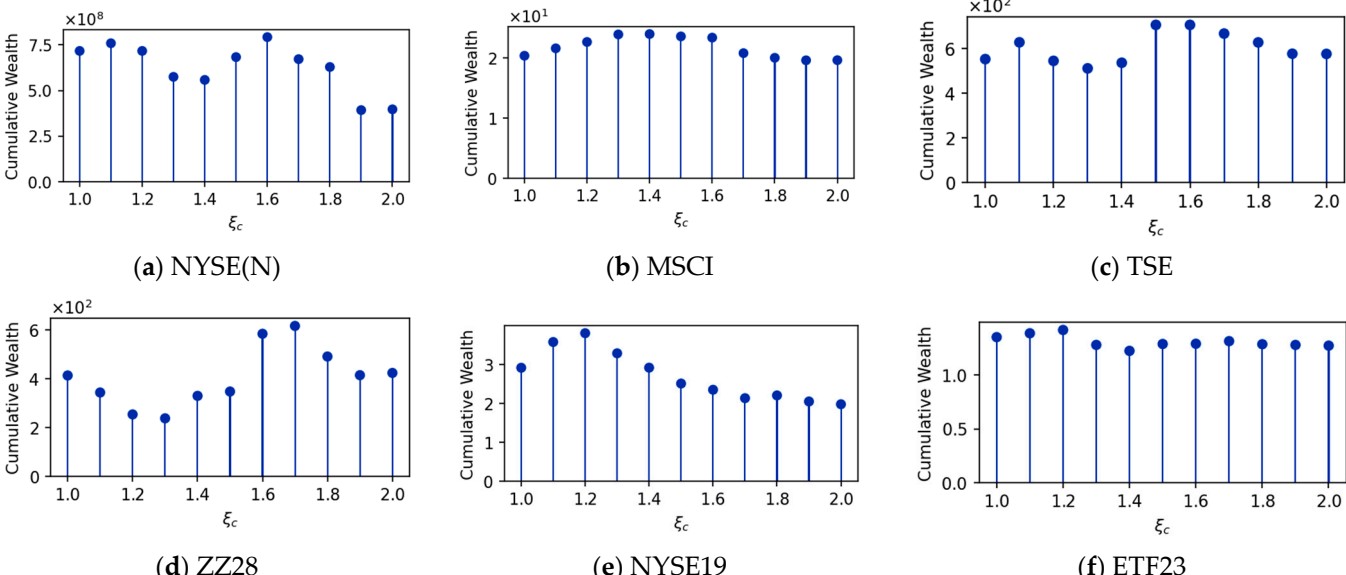

**Figure 4.** Fixed $\epsilon = 30$, the cumulative wealth changes with the variation of $\xi_c$ on six datasets for PAE-C.

### 4.4.2. Ensemble Effectiveness

To assess the effectiveness of the PAE framework in combining diverse trends to improve investment performance, we compare its performance against a single trend situation and analyze the distribution of the weight vector.

Different trend estimators exhibit significant variability in performance across various datasets. As we initially emphasized, it is impossible to predict which trend estimator will demonstrate the best performance over the entire investment period. Therefore, we evaluate the effectiveness of our ensemble strategy from two perspectives. First, we expect the strategy to improve performance. Second, the baseline for evaluating this strategy is that its performance should not be inferior to that of any single trend estimator.

In Table 3, SMA, EMA, IP, and PP mean the corresponding weight in the weight vector **w** is 1, and the others are 0 (for example, SMA means **w** = [1, 0, 0, 0]) in the entire investment period. The best performing strategy is highlighted in bold, while the two

worst performing strategies are underlined. The results from Table 3 clearly show that our PAE-C and PAE-R strategies consistently exceed the baseline method. Furthermore, they demonstrate leading performance on most datasets. It significantly validates the effectiveness of the PAE strategy in integrating trend estimators.

**Table 3.** Cumulative wealth of different trend estimators. The best performing strategy is highlighted in bold, while the two worst performing strategies are underlined. PP, EMA, SMA, and IP mean the corresponding weight in the weight vector **w** is 1, and the others are 0.

| Trend Estimator | NYSE(N) | MSCI | TSE | ZZ28 | NYSE19 | ETF23 |
|---|---|---|---|---|---|---|
| PP | $2.08 \times 10^9$ | 8.33 | 226.84 | **906.7** | 1.46 | 1.4 |
| EMA | $4.64 \times 10^8$ | 23.6 | 680.83 | 283.58 | 2.46 | 1.69 |
| SMA | $4.26 \times 10^8$ | 14.1 | 76.77 | 134.64 | 1.14 | 1.26 |
| IP | $1.16 \times 10^6$ | 10.28 | $1.39 \times 10^3$ | 185.13 | **9.34** | 0.84 |
| PAE-C | $6.83 \times 10^8$ | **23.63** | 706 | 348.31 | 2.51 | 1.29 |
| PAE-R | **$4.15 \times 10^9$** | 14.98 | **$2.26 \times 10^3$** | 867.83 | 2.44 | **1.71** |

In addition to the cumulative wealth of different trend estimators, Figure 5 illustrates the evolution of the weighting vector during the entire investment on ZZ28. In most times, the PAE-R and PAE-C frameworks are able to balance each trend estimator. To be specific, from time 0 to 3000, we can observe that the investment performance of IP is poor compared with the other three trends. In the case of PAE-C, the IP has less weight and ultimately decreases to 0, while increasing the weights of EMA and PP leads to better investment performance. In the entire investment, the average respective proportions of SMA, EMA, IP, and PP are 6.3%, 71.4%, 3.0%, and 19.3%.

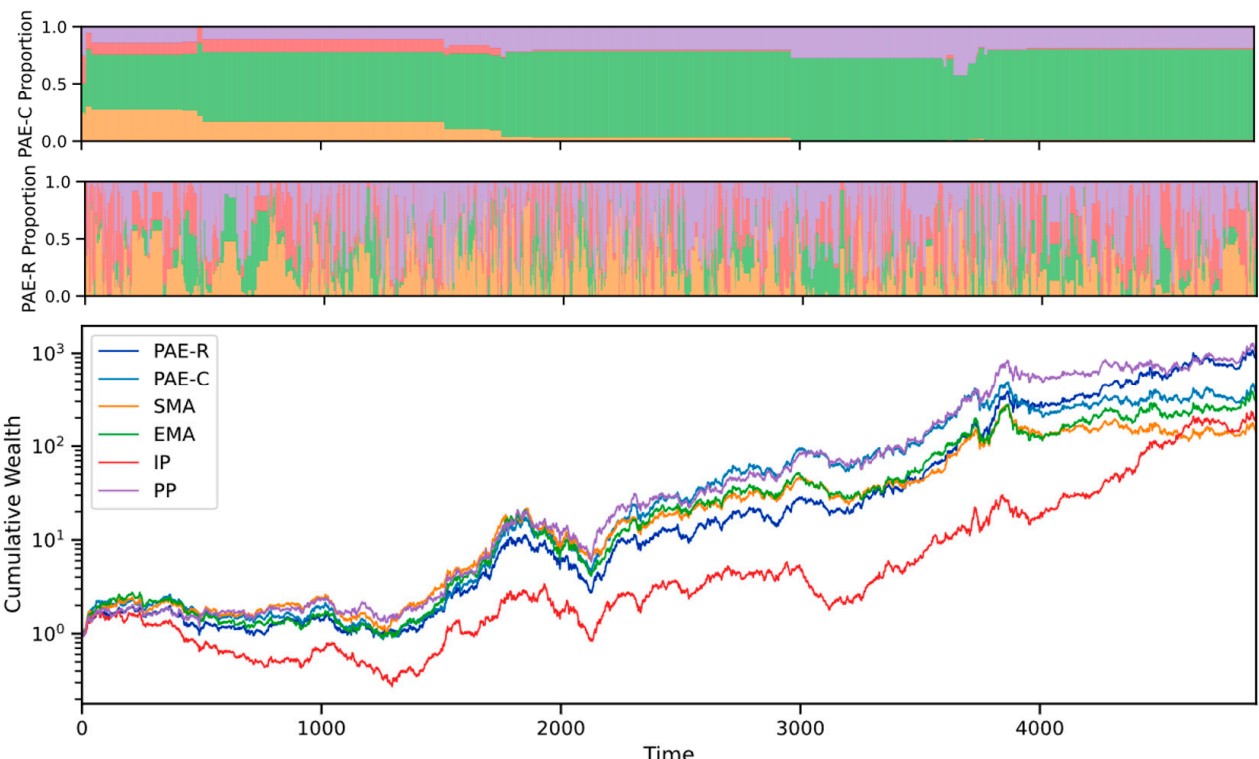

**Figure 5.** Cumulative wealth of different trend estimators and the distribution of the weighting vector during the entire investment on ZZ28. These images share the same x-axis and legend.

Compared with PAE-C, PAE-R is better at capturing changes in short-term investment performance across different trend estimators and is more aggressive in adjusting weights.

From time 0 to 3000, the average proportions of SMA, EMA, IP, and PP are 25.4%, 17.3%, 25.1%, and 32.2%. But, from time 4000 to the end, IP performed very well, and its investment performance even caught up with SMA. During this period, the average respective proportions of SMA, EMA, IP, and PP are 17.9%, 17.6%, 30.3%, and 34.3%. The decrease in the proportion of SMA and the increase in the proportion of IP indicate that PAE-R can integrate different trend estimators.

### 4.4.3. Comparison Studies

Table 4 shows the performance of PAE and competing strategies on six real-world datasets. We evaluate the portfolio strategies comprehensively with respect to return and risk-adjusted return.

**Table 4.** Performance of different strategies. The best performing strategy for each row is highlighted in bold.

| Dataset | Metrics | BAH | OLMAR | AICTR | SPOLC | TPPT | PAE-C | PAE-R |
|---------|---------|-----|-------|-------|-------|------|-------|-------|
| NYSE(N) | CW | 18.29 | $4.19 \times 10^8$ | $1.01 \times 10^9$ | $1.99 \times 10^7$ | $2.63 \times 10^9$ | $6.83 \times 10^8$ | $\mathbf{4.15 \times 10^9}$ |
|  | APY | 0.121 | 1.177 | 1.254 | 0.932 | 1.339 | 1.219 | **1.382** |
|  | SR | 0.046 | 0.104 | 0.106 | 0.105 | 0.108 | 0.105 | **0.113** |
|  | IR | −0.025 | 0.096 | 0.099 | 0.095 | 0.102 | 0.097 | **0.107** |
|  | CR | 0.225 | 1.28 | 1.374 | 1.082 | 1.561 | 1.298 | **1.629** |
| MSCI | CW | 0.89 | 14.5 | 12.38 | 7.34 | 10.81 | **23.63** | 14.98 |
|  | APY | −0.027 | 0.908 | 0.837 | 0.618 | 0.776 | **1.147** | 0.923 |
|  | SR | 0.001 | 0.116 | 0.108 | 0.09 | 0.103 | **0.132** | 0.116 |
|  | IR | −0.036 | 0.169 | 0.158 | 0.132 | 0.155 | **0.193** | 0.17 |
|  | CR | −0.041 | 1.889 | 2.026 | 1.13 | 1.483 | **2.677** | 1.901 |
| TSE | CW | 1.56 | 57.79 | 544.47 | 277.1 | 265.10 | 706 | $\mathbf{2.26 \times 10^3}$ |
|  | APY | 0.093 | 1.252 | 2.529 | 2.083 | 2.055 | 2.717 | **3.692** |
|  | SR | 0.048 | 0.082 | 0.111 | 0.111 | 0.101 | 0.114 | **0.129** |
|  | IR | −0.002 | 0.078 | 0.108 | 0.107 | 0.098 | 0.111 | **0.127** |
|  | CR | 0.311 | 1.527 | 3.81 | 4.087 | 2.672 | 4.779 | **5.127** |
| ZZ28 | CW | 31.76 | 124.58 | 195.73 | 853.07 | 123.5 | 348.31 | **867.83** |
|  | APY | 0.194 | 0.281 | 0.311 | 0.415 | 0.280 | 0.351 | **0.416** |
|  | SR | 0.048 | 0.05 | 0.053 | **0.068** | 0.050 | 0.057 | 0.063 |
|  | IR | 0.002 | 0.024 | 0.029 | **0.044** | 0.024 | 0.034 | 0.043 |
|  | CR | 0.335 | 0.389 | 0.436 | **0.711** | 0.430 | 0.474 | 0.548 |
| NYSE19 | CW | 1.37 | 0.98 | 1.79 | 2.45 | 1.56 | **2.51** | 2.44 |
|  | APY | 0.07 | −0.005 | 0.133 | 0.214 | 0.101 | **0.22** | 0.213 |
|  | SR | 0.034 | 0.018 | 0.032 | **0.04** | 0.029 | **0.04** | 0.039 |
|  | IR | −0.022 | 0.009 | 0.024 | **0.033** | 0.021 | **0.033** | 0.032 |
|  | CR | 0.288 | −0.006 | 0.168 | 0.299 | 0.128 | 0.277 | **0.301** |
| ETF23 | CW | 0.87 | 1.19 | 1.28 | 0.84 | 1.07 | 1.29 | **1.71** |
|  | APY | −0.053 | 0.07 | 0.102 | −0.064 | 0.027 | 0.104 | **0.231** |
|  | SR | −0.012 | 0.023 | 0.028 | −0.003 | 0.016 | 0.029 | **0.048** |
|  | IR | −0.005 | 0.036 | 0.043 | 0.005 | 0.028 | 0.043 | **0.067** |
|  | CR | −0.185 | 0.156 | 0.24 | −0.181 | 0.057 | 0.224 | **0.532** |

In terms of return, PAE-R demonstrates the best performance for most datasets, especially in TSE and NYSE(N), where its CW far exceeds that of other strategies. PAE-C performs well in the MSCI and ETF23 datasets but is generally inferior to PAE-R for the remaining datasets. To provide a visual representation of the trajectory's evolution, CW plots are presented in Figure 6, revealing that the PAE algorithm consistently attains higher wealth returns compared to others over the majority of time periods, demonstrating the superiority of the model in portfolio decision-making. Regarding risk-adjusted returns, PAE-R outperforms other strategies for most datasets, especially in ETF23 and TSE. PAE-C exhibits advantages on specific datasets, such as MSCI and NYSE19. Notably, SPOLC,

optimized for maximum drawdown, demonstrates comparable performance to PAE in terms of risk-adjusted returns. PAE-C shows strength in the MSCI and NYSE19 datasets but is generally inferior to PAE-R for other datasets.

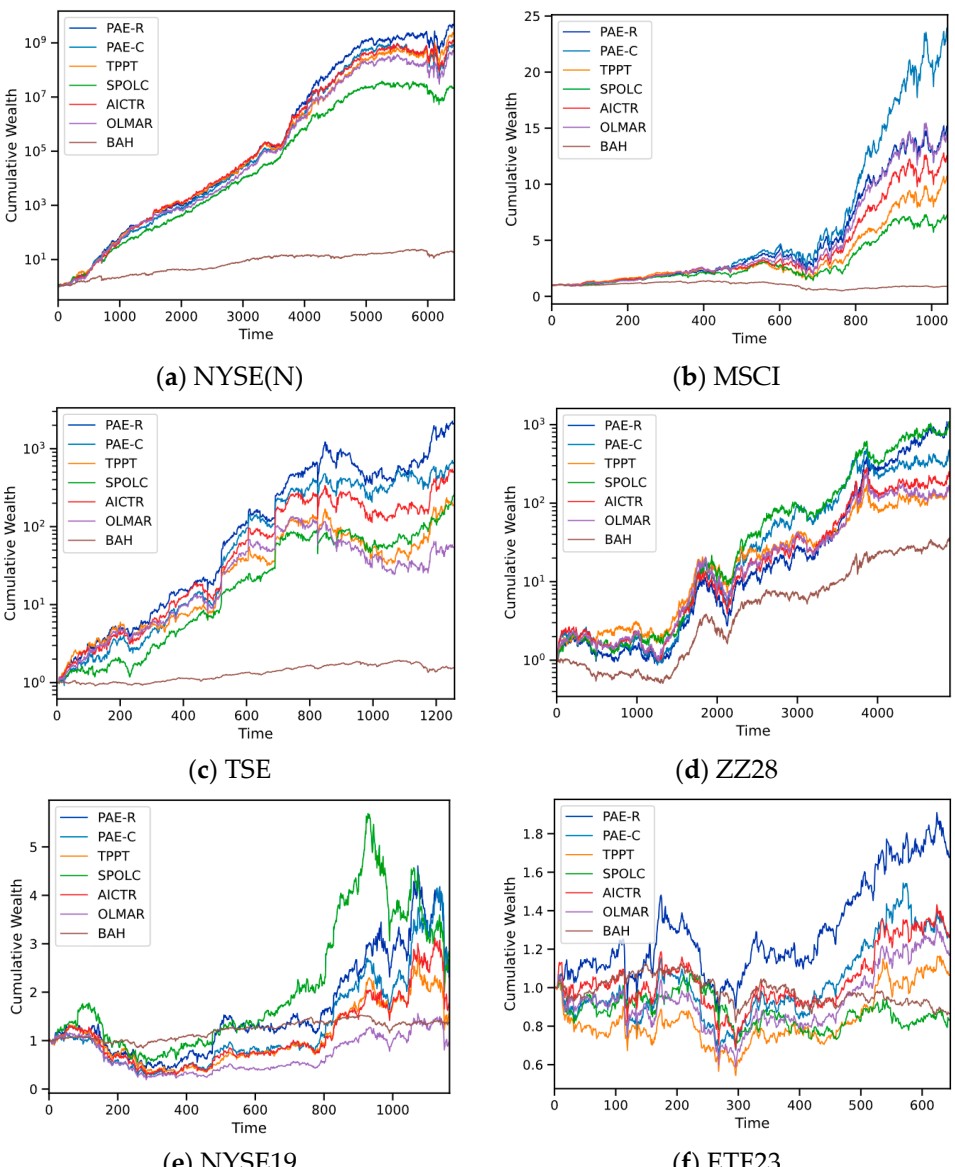

**Figure 6.** Cumulative wealth of different portfolio selection strategies during the entire investments on six datasets.

In addition to the quantitative evaluation results, Table 5 lists the results of the *t*-test to justify the effectiveness of the proposed PAE. Table 5 shows that except for NYSE19 and ETF23, it is almost impossible for PAE-C to produce the corresponding returns simply by luck at a high confidence level of 97%. There is also around an 80% confidence level for PAE-C not to produce their returns by luck in the other datasets. The results also indicate that the values of $\alpha$ for PAE-R are significantly larger than 0 at a high confidence level of 99% on four datasets, while the rest are higher than 90%.

In general, PAE shows noticeable advantages over other competing strategies in terms of comprehensive performance.

**Table 5.** Statistical *t*-test of the performance of PAE-C and PAE-R. MER stands for mean excess return.

| Statistics | NYSE(N) PAE-C | NYSE(N) PAE-R | MSCI PAE-C | MSCI PAE-R | TSE PAE-C | TSE PAE-R | ZZ28 PAE-C | ZZ28 PAE-R | NYSE19 PAE-C | NYSE19 PAE-R | ETF23 PAE-C | ETF23 PAE-R |
|---|---|---|---|---|---|---|---|---|---|---|---|---|
| MER | 0.0032 | 0.0035 | 0.0033 | 0.0030 | 0.0066 | 0.0076 | 0.0007 | 0.0008 | 0.0011 | 0.0011 | 0.0010 | 0.0012 |
| MER-market | 0.0006 | / | 0.0000 | / | 0.0004 | / | 0.0009 | / | 0.0004 | / | −0.0001 | / |
| $\alpha$ | 0.0030 | 0.0033 | 0.0033 | 0.0029 | 0.0061 | 0.0071 | 0.0007 | 0.0009 | 0.0009 | 0.0014 | 0.0008 | 0.0012 |
| $\beta$ | 1.3411 | 1.3561 | 1.2009 | 1.1946 | 2.2057 | 2.0434 | 1.0666 | 1.0718 | 1.7696 | 1.4239 | 1.1315 | 1.1495 |
| *t*-statistics | 7.3420 | 8.0863 | 6.3287 | 5.5531 | 3.6954 | 4.2855 | 2.2146 | 2.8055 | 1.3519 | 1.7481 | 1.1124 | 1.7443 |
| *p*-value | 0.0000 | 0.0000 | 0.0000 | 0.0000 | 0.0002 | 0.0000 | 0.0268 | 0.0050 | 0.1769 | 0.0807 | 0.2664 | 0.0816 |

#### 4.4.4. Transaction Costs

The paramount concern in practical real-world trading is the transaction cost. When there is a transaction cost rate of *r* for each trade in the portfolio re-balancing process, the CW can be determined using the proportional transaction cost model [9] as follows:

$$S_n^r = S_0 \prod_{t=1}^{n} \left[ \left( \hat{\mathbf{b}}_t^\top \mathbf{x}_t \right) \times \left( 1 - \frac{r}{2} \sum_{i=1}^{m} \left| \hat{\mathbf{b}}_t^{(i)} - \overset{\sim(i)}{\mathbf{b}}_{t-1} \right| \right) \right], \tag{30}$$

where $\overset{\sim(i)}{\mathbf{b}}_{t-1} = \frac{\hat{\mathbf{b}}_{t-1}^{(i)} * \mathbf{x}_{t-1}^{(i)}}{\hat{\mathbf{b}}_{t-1}^\top \mathbf{x}_{t-1}}$ is the price adjusted portfolio of asset *i* in the *t*th period and $\mathbf{b}_0$ is

set to $[0, \dots, 0]^\top$. The term $(r/2)\sum_{i=1}^{m} \left| \hat{\mathbf{b}}_t^{(i)} - \overset{\sim(i)}{\mathbf{b}}_{t-1} \right|$ represents the transaction cost incurred

from the adjustment of portfolio $\overset{\sim}{\mathbf{b}}_{t-1}$ to $\mathbf{b}_t$ through re-balancing.

To evaluate the practicality of the portfolio selection strategies, we perform experiments on CW while changing the transaction cost rate *r* between 0 and 0.5%. The findings, displayed in Figure 7, indicate that PAE-R delivers superior results on NYSE(N), MSCI, TSE, and ETF23. Additionally, PAE-R shows comparable performance to other state-of-the-art strategies on the remaining datasets, excelling in managing transaction costs. PAE-C also demonstrates competitive performance, particularly in MSCI and NYSE19.

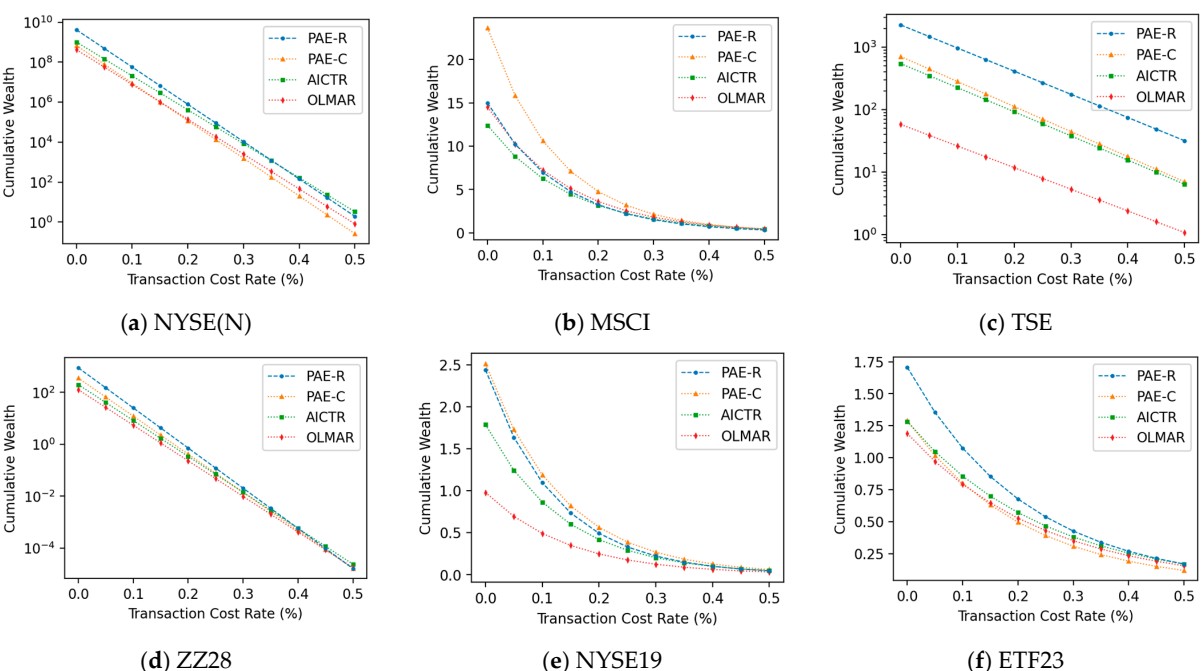

**Figure 7.** Cumulative wealth of different strategies under different transaction cost rates on six benchmark datasets.

This underscores the capability of PAE as a robust framework for managing transaction costs, making it well-suited for real-world financial environments.

### 4.4.5. Running Time

To evaluate the computational cost of the PAE, we conducted 5 iterations of the algorithms on each dataset using a Ryzen 5 3500X CPU, which was designed by AMD in the USA, paired with two 16GB DDR4 3200 MHz RAM modules.. The average running times (in seconds) of PAE-C per trading period are 0.0064, 0.0064, 0.0071, 0.0064, 0.0067, and 0.0073 on NYSE(N), MSCI, TSE, ZZ28, NYSE19, and ETF23, respectively. The average running times (in seconds) of PAE-R per trading period are 0.0056, 0.0056, 0.0063, 0.0057, 0.0056, and 0.0060. The average running times (in seconds) of TPPT per trading period are 0.0092, 0.0099, 0.0117, 0.0098, 0.0090, and 0.0111. The average running times (in seconds) of AICTR per trading period are 0.0149, 0.0145, 0.0145, 0.0141, 0.0150, and 0.0172. In comparison, TPPT and AICTR exhibit higher average running times per trading period, suggesting that PAE demonstrates efficient computations suitable for handling multiple trend estimators.

## 5. Conclusions and Future Work

In this study, we present a passive aggressive ensemble (PAE) framework, a new approach for online portfolio selection that integrates multiple trend estimators. PAE stands out by efficiently combining different estimators and enhancing their performance through a novel weighting mechanism. Our extensive experiments across various real-world datasets demonstrate that PAE not only outperforms the competing algorithms in key evaluation metrics but also shows potential for managing transaction costs effectively. This approach is particularly notable for its adaptability to different market conditions and scalability, making it suitable for practical financial applications.

This paper envisages three primary directions for future research. First, the exploration of different loss functions: In our framework, the weighting factors of different trend estimators are determined by their relative loss to the market. Identifying more optimal loss functions to evaluate the effectiveness of trend estimators could significantly enhance the overall performance of the framework. Second, the investigation into the application of regularization techniques in ensemble methods: Regularization techniques can prevent overfitting and bolster the robustness of the ensemble approach. Finally, we acknowledge that our model has substantial room for improvement in high transaction fee environments. Therefore, refining the PAE model to improve its performance by considering transaction costs is also a pivotal aspect of our forthcoming research endeavors.

**Author Contributions:** Conceptualization, K.X.; methodology, K.X.; software, K.X.; validation, K.X.; formal analysis, K.X.; investigation, K.X.; resources, K.X.; data curation, K.X.; writing—original draft preparation, K.X.; writing—review and editing, H.Y. and Y.C.; visualization, K.X.; supervision, J.Y., H.Y., H.F. and Y.C.; project administration, H.F. and Y.C.; funding acquisition, Y.C. All authors have read and agreed to the published version of the manuscript.

**Funding:** This research was funded by the Stabilization Support Plan for Shenzhen Higher Education Institutions, grant number 20200812165210001.

**Data Availability Statement:** The dataset, i.e., ETF23, is free to download from the website: https://drive.google.com/file/d/1W89bkqKqQJioLC9ljIf_mlFHgk9fPcSh/view?usp=sharing, or https://pan.baidu.com/s/1j3wMPVHHWiifKW4jJ2iS-Q?pwd=ETF0 (accessed on 2 March 2024).

**Conflicts of Interest:** The authors declare no conflicts of interest.

## Appendix A

**Proof of Theorem 1.** When $\ell_{\xi_r} = 0$, the constraint in (16) is satisfied by $\mathbf{w}_t$, and it becomes the optimal solution. For the case where $\ell_{\xi_r} \neq 0$, we can solve the optimization problem by introducing the Lagrangian of the problem in (16):

$$\mathscr{L}(\mathbf{w}, \tau_{1,t}, \lambda) = \frac{1}{2}\| \mathbf{w} - \mathbf{w}_t \|^2 + \tau_{1,t}\left(\mathbf{r}_{*,t} - \mathbf{w}^\top \mathbf{r}_t - \xi_r\right) + \lambda\left(\mathbf{1}^\top \mathbf{w} - 1\right), \qquad \text{(A1)}$$

where $\tau_{1,t} \geq 0$ and $\lambda$ are the Lagrangian multipliers. Taking the gradient with respect to $\mathbf{w}$ and setting it to zero, we obtain the following:

$$\frac{\partial \mathscr{L}}{\partial \mathbf{w}} = (\mathbf{w} - \mathbf{w}_t) - \tau_{1,t}\mathbf{r}_t + \lambda\mathbf{1} = 0. \qquad \text{(A2)}$$

By left multiplying $\mathbf{1}^\top$ on both sides, we obtain the following:

$$1 = 1 - \tau_{1,t}\mathbf{1}^\top \mathbf{r}_t + \lambda L, \qquad \text{(A3)}$$

$$\lambda = \tau_{1,t}\bar{r}_t, \qquad \text{(A4)}$$

where $\bar{r}_t = \frac{\mathbf{1}^\top \mathbf{r}_t}{L}$ is the mean value of back-tested returns. Plugging the above equation to (A2), we obtain the update of $\mathbf{w}$ as follows:

$$\mathbf{w} = \mathbf{w}_t + \tau_{1,t}(\mathbf{r}_t - \bar{r}_t\mathbf{1}). \qquad \text{(A5)}$$

Simplifying the formula after plugging (A4) and (A5) to Lagrangian (A1), we obtain the following:

$$\mathscr{L}(\tau_{1,t}) = -\frac{1}{2}\tau_{1,t}^2\| \mathbf{r}_t - \bar{r}_t\mathbf{1} \|^2 + \tau_{1,t}\left(r_{*,t} - \mathbf{w}_t^\top \mathbf{r}_t - \xi_r\right). \qquad \text{(A6)}$$

Taking derivative with respect to $\tau_{1,t}$ and setting it to zero, we have the following:

$$\frac{\partial \mathscr{L}}{\partial \tau_{1,t}} = -\tau_{1,t}\| \mathbf{r}_t - \bar{r}_t\mathbf{1} \|^2 + \left(r_{*,t} - \mathbf{w}_t^\top \mathbf{r}_t - \xi_r\right) = 0 \quad, \qquad \text{(A7)}$$

which implies the following:

$$\tau_{1,t} = \frac{r_{*,t} - \mathbf{w}_t^\top \mathbf{r}_t - \xi_r}{\| \mathbf{r}_t - \bar{r}_t\mathbf{1} \|^2}. \qquad \text{(A8)}$$

Further projecting $\tau_{1,t}$ to $[0, +\infty)$, we complete the proof of (19). $\square$

## Appendix B

**Proof of Theorem 3.** The Lagrangian of the optimization problem (17) is as follows:

$$\mathscr{L}(\mathbf{b}, \tau_{2,t}, \eta) = \frac{1}{2}\| \mathbf{b} - \mathbf{b}_t \|^2 + \tau_{2,t}(\epsilon - \mathbf{b}^\top \hat{\mathbf{X}}_{t+1}\mathbf{w}_{t+1}) + \eta(\mathbf{1}^\top \mathbf{b} - 1), \qquad \text{(A9)}$$

where $\tau_{2,t} \geq 0$ and $\eta$ are the Lagrangian multipliers. Taking the gradient with respect to $\mathbf{b}$ and setting it to zero, we have the following:

$$\frac{\partial \mathscr{L}}{\partial \mathbf{b}} = (\mathbf{b} - \mathbf{b}_t) - \tau_{2,t}\hat{\mathbf{X}}_{t+1}\mathbf{w}_{t+1} + \eta\mathbf{1}_{(m)} = 0, \qquad \text{(A10)}$$

where $\mathbf{1}_{(m)}$ denotes the $m$-dimensional $\mathbf{1}$ vector. By left multiplying both sides with $\mathbf{1}_{(m)}^{\top}$, we have the following:

$$1 = 1 - \tau_{2,t}\mathbf{1}_{(m)}^{\top}\hat{\mathbf{X}}_{t+1}\mathbf{w}_{t+1} + \eta m, \tag{A11}$$

$$\eta = \tau_{2,t}\mu, \tag{A12}$$

where $\mu = \frac{1}{m}\mathbf{1}_{(m)}^{\top}\hat{\mathbf{X}}_{t+1}\mathbf{w}_{t+1}$. Plugging the above equation to (A10), we have the update of $\mathbf{b}$ as follows:

$$\mathbf{b} = \mathbf{b}_t + \tau_{2,t}\Big(\mathbf{b}_t - \mu\mathbf{1}_{(m)}\Big). \tag{A13}$$

Simplifying the formula after plugging (A12) and (A13) to Lagrangian (A9), we have the following:

$$\mathscr{L}(\tau_{2,t}) = -\frac{1}{2}\tau_{2,t}^2\|\mathbf{b}_t - \mu\mathbf{1}\|^2 + \tau_{2,t}\Big(\epsilon - \mathbf{b}^{\top}\hat{\mathbf{X}}_{t+1}\mathbf{w}_{t+1}\Big). \tag{A14}$$

Taking the derivative with respect to $\tau_{2,t}$ and setting it to zero, we arrive at the following:

$$\frac{\partial\mathscr{L}}{\partial\tau_{2,t}} = -\tau_{2,t}\|\mathbf{b}_t - \mu\mathbf{1}\|^2 + \Big(\epsilon - \mathbf{b}_t^{\top}\hat{\mathbf{X}}_{t+1}\mathbf{w}_{t+1}\Big) = 0, \tag{A15}$$

which can be simplified to the following:

$$\tau_{2,t} = \frac{\epsilon - \mathbf{b}_t^{\top}\hat{\mathbf{X}}_{t+1}\mathbf{w}_{t+1}}{\|\hat{\mathbf{X}}_{t+1}\mathbf{w}_{t+1} - \mu\mathbf{1}_{(m)}\|^2}. \tag{A16}$$

Further projecting $\tau_{2,t}$ to $[0, +\infty)$, we finally complete the proof of (24). □

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
