# Peer review of "Passive Aggressive Ensemble for Online Portfolio Selection"

_mathematics, doi:10.3390/math12070956_

Round 1

Reviewer 1 Report

Comments and Suggestions for Authors

How do the contributions distinguish themselves from related work? Unclear.

Utilizing a strategy to pick hyperparameters appears to have limited experimental value rather than providing a robust scientific generalization of an issue.

Is it customary in journal formatting for the problem to have a separate section before the related work? The current arrangement does not facilitate readers in comprehending the significance and actual deficiencies in the text.

I would appreciate having access to the descriptive statistics of the data gathered and utilized in the trials.

Instead of only providing descriptions of figures and tables (results section), it would be beneficial to have an in-depth analysis and insights into the differences in procedures and outputs.

All references are over 4 years old. Consulting current literature might help in choosing methodologies and offer fresh insights into analyzing the problem scenarios.

Comments on the Quality of English Language

English is okay, it would be nice to do a last copy-edit review especially to make things standard (e.g., acronyms, references-link, etc.).

Reviewer 2 Report

Comments and Suggestions for Authors

This paper addresses the problem of predetermining trends in finance times series to maximize the final cumulative wealth for online portfolio selection tasks.

According with authors, the viable solution to address such problem is to ensemble learning algorithm to amalgamate various trend estimators emerges. Therefore, they tackle the problem of online portfolio selection with multiple trend estimators by developing a framework named Passive Aggressive Ensemble (PAE) which combines the passive aggressive principle of Cramer et al (2013) and the online ensemble framework of Von Krannichfeldt et al. (2020). In this paper, two loss functions are proposed to combine all the trends. The PAE-R and the PAE-C strategies allows for different trends and consequently adapts to different financial environments.

The authors demonstrate their algorithm outperforms other strategies evaluating different financial criteria’s in different financial datasets except ZZ28.

I would recommend the authors to write a deep analysis of the results and explain the reason about this particular result, and in terms of optimization, are you finding the global optimum solution? if not what far are your results from the global optimum? and how fast is your algorithm in comparison with the other strategies to find a feasible if not the global solution?

Comments on the Quality of English Language

You have few grammatical errors that must be fixed, but english is good.

Round 2

Reviewer 1 Report

Comments and Suggestions for Authors

N/A

Comments on the Quality of English Language

Tense consistency, and acronyms should still be reviewed.